# Prevalence of Hyperuricemia and Its Association with Cardiovascular Risk Factors and Subclinical Target Organ Damage

**DOI:** 10.3390/jcm12010050

**Published:** 2022-12-21

**Authors:** Paula Antelo-Pais, Miguel Ángel Prieto-Díaz, Rafael M. Micó-Pérez, Vicente Pallarés-Carratalá, Sonsoles Velilla-Zancada, José Polo-García, Alfonso Barquilla-García, Leovigildo Ginel-Mendoza, Antonio Segura-Fragoso, Facundo Vitelli-Storelli, Vicente Martín-Sánchez, Álvaro Hermida-Ameijerias, Sergio Cinza-Sanjurjo

**Affiliations:** 1Santa Comba Health Centre, Health Area of Santiago de Compostela, 15970 Santiago de Compostela, Spain; 2Vallobín-La Florida Health Centre, 33012 Oviedo, Spain; 3Fontanars dels Alforins Health Centre, Xàtiva–Ontinyent Department of Health, 46635 Valencia, Spain; 4Health Surveillance Unit, Mutual Insurance Union, 12004 Castellon, Spain; 5Department of Medicine, Jaume I University, 12071 Castellon, Spain; 6Joaquin Elizalde Health Centre, 26004 Logroño, Spain; 7Casar de Cáceres Health Centre, 10200 Cáceres, Spain; 8Trujillo Health Center, 10200 Cáceres, Spain; 9Ciudad Jardín Health Center, 29014 Málaga, Spain; 10Epidemiology Unit, Semergen Research Agency, 28029 Madrid, Spain; 11Gene-Environment-Health Interaction Research Group (GIIGAS)/Institute of Biomedicine (IBIOMED), University of León, 24071 León, Spain; 12Institute of Biomedicine (IBIOMED), Epidemiology and Public Health Networking Biomedical Research Centre (CIBERESP), University of León, 24071 León, Spain; 13Department of Internal Medicine, University Hospital of Santiago de Compostela, 15706 A Coruña, Spain; 14Porto do Son Health Centre, Health Area of Santiago de Compostela, Health Research Institute of Santiago de Compostela (IDIS), Networking Biomedical Research Centre-Cardiovascular Diseases (CIBERCV), 15970 Santiago de Compostela, Spain

**Keywords:** hyperuricemia, cardiovascular risk, prevalence, associated risk

## Abstract

The role of uric acid levels in the cardiovascular continuum is not clear. Our objective is to analyze the prevalence of hyperuricemia (HU) and its association with cardiovascular risk factors (CVRF), subclinical target organ damage (sTOD), and cardiovascular diseases (CVD). We evaluated the prevalence of HU in 6.927 patients included in the baseline visit of the IBERICAN study. HU was defined as uric acid levels above 6 mg/dL in women, and 7 mg/dL in men. Using adjusted logistic regression models, the odds ratios were estimated according to CVRF, sTOD, and CVD. The prevalence of HU was 16.3%. The risk of HU was higher in patients with pathological glomerular filtration rate (aOR: 2.92), heart failure (HF) (aOR: 1.91), abdominal obesity (aOR: 1.80), hypertension (HTN) (aOR: 1.65), use of thiazides (aOR: 1.54), left ventricular hypertrophy (LVH) (aOR: 1.36), atrial fibrillation (AFIB) (aOR: 1.29), and albuminuria (aOR: 1.27). On the other hand, being female (aOR: 0.82) showed a reduced risk. The prevalence of HU was higher in men, in patients presenting CVRF such as HTN and abdominal obesity, and with co-existence of LVH, atrial fibrillation (AFIB), HF, and any form of kidney injury. These associations raise the possibility that HU forms part of the early stages of the cardiovascular continuum. This may influence its management in Primary Healthcare because the presence of HU could mean an increased CV risk in the patients.

## 1. Introduction

Cardiovascular diseases (CVD) are the leading cause of death in Spain, particularly in women [1], and are also the second leading cause of years of life lost after cancer [2]. A decreasing trend, in standardized rates, has been observed in the last few decades which is due to the improvements in primary prevention, lifestyle modification, better control of cardiovascular risk factors (CVRF), and better management and treatment, mainly interventional, in the acute phase of events [3]. However, despite the better control of CVRF, many patients develop CVD, suggesting that there may be remaining factors which are not controlled.

Atherosclerosis is the pathophysiological base of CVD, which progresses silently over decades due to the effect of CVRF and of subclinical target organ damages (sTOD) to the development of CVD [4]. This is what is known as the cardiovascular continuum.

Among CVRF, the potential role of uric acid—a product of xanthine oxidase—in the development of CVD has been set forth, since high uric acid levels have been identified in the context of oxidative stress and of CVD, such as ischemic heart disease (IHD) and heart failure (HF) [5].

Hyperuricemia (HU) is associated with a state of oxidative stress that produces endothelial dysfunction, increases oxidation of LDL, and promotes a proinflammatory state which contributes to the development of atherosclerosis and its thrombotic complications [6,7,8,9]. The presence of uric acid in the atheromatous plaque is the reason why the classical cut-off points defining HU (6 mg/dL in women, and 7 mg/dL in men) are based on the levels of precipitation in uric acid [10,11].

At present, there are studies available that provide solid results on the relationship between HU and CVRF such as hypertension (HTN) [12], diabetes mellitus (DM) [13], obesity [14], hypercholesterolemia [15], and metabolic syndrome (MetS) [13,16], as well as between HU and sTOD [17,18]. Nevertheless, results are more controversial and contradictory regarding its association with CVD [19,20,21]. These discrepancies could be justified by the presence of uncontrolled confounding factors in those studies, for example, the difference between the sexes, the peripheral insulin resistance, or the MetS [22]. The real role of HU and how it is associated with other CVRF has remained unknown until now. A relationship between HU and several metabolic conditions (DM, MetS, obesity, or HTN) has been suggested. In addition, some authors explain the association between HU and a worse prognosis in patients with an acute coronary disease [23] or after the initial hospitalization for HF [24].

Forming a complete picture about patients and the CVRF, the risk factors needing to be controlled, and the sTOD or CVD rates would allow us to control the confounding effect among all the variables and to further the study of the role of HU in the cardiovascular continuum, as is the case of the IBERICAN study (Identification of Spanish Population in Cardiovascular and Renal Risk) [25].

The objective of this study is to analyze which CVRF, sTOD, and CVD are associated with HU in patients seen in primary care practices.

## 2. Materials and Methods

### 2.1. Study Design

Cross-sectional HU prevalence study conducted in patients included in the IBERICAN study [25].

The study was approved by the Ethics Committee for the investigation with medicinal products (CEIm) of the Hospital Clínico San Carlos of Madrid on 21 February 2013 (C.P. IBERICAN-C.I. 13/047-E) and is registered at https://clinicaltrials.gov with the number NCT02261441 (accessed on 1 November 2022).

### 2.2. Selection of Patients

Patients aged between 18 and 85 were recruited consecutively in primary care practices with the following inclusion criteria: (1) user of the National Health System, (2) residing in Spain during the last 5 years, (3) registered with the physician researcher, (4) signing the informed consent form. The exclusion criteria were (1) change of habitual residence to another town or country in the next 6 months, (2) terminal illness or reduced life expectancy in the next 5 years, (3) clear difficulty in being followed up in PC, or (4) refusal to form part of the cohort in the first place or to continue to be part of it during the follow-up. The estimated sample size has been detailed in previous publications [25].

### 2.3. Variables Recorded

Socio-demographic data of each patient were recorded in the entry visit (sex, age, habitat, level of education, family economic status, and current employment situation), as well as toxic habits (tobacco and alcohol consumption), family history of early CVD and personal history (HTN, DM, hypercholesterolemia, atrial fibrillation—AFIB-, HF, peripheral artery disease, or cerebrovascular disease), clinical parameters (weight, height, body mass index—BMI-, waist circumference, systolic pressure, diastolic pressure, pulse pressure—PP-, and heart rate), information on the presence or absence of each CVRF as well as their treatment. The complementary tests recorded were lab tests (blood count, blood chemistry, and urine test) and electrocardiogram, which were considered valid if performed within the last 6 months before the patient’s inclusion. The equation CKD-EPI was used to calculate the glomerular filtration rate (eGFR) [26].

The HU was defined as uric acid levels above 6 mg/dL in women and 7 mg/dL in men, as set out in the guidelines on HTN of the European societies 2018 [27]. All other variables have been described in previous publications [25], and the CVRF together with their degree of control have been defined according to the respective clinical practice guidelines [21]. The term CVD includes here IHD, HF, peripheral artery disease, and cerebrovascular disease, as set out in the clinical practice guidelines [27].

### 2.4. Statistical Analysis

Qualitative variables have been defined as percentages with a 95% confidence interval (95% CI). For quantitative variables, the Shapiro–Wilk test has been used to check the data fitting to normal distribution. If the variable showed normal distribution, it has been described using the arithmetic mean and the standard deviation (SD); otherwise, the median and the interquartile range have been used.

The bivariate analysis has been performed using chi-square for qualitative variables, and ANOVA for quantitative variables, comparing patients with HU with patients without HU.

Unconditional logistic regression models were used to estimate the odds ratios adjusting (aOR), as the case may be, for socio-demographic variables (sex, age, level of education, habitat, employment, income, and race) or clinical variables. Results are presented with their 95% confidence intervals (95% CI). Finally, a multivariate analysis was carried out to study the relationship of CVRF and demographic variables with HU. All analyses were performed with the program STATA version 16. We consider statistically significant a *p*-value < 0.05.

## 3. Results

The study included the 6927 patients with available data on uric acid levels. The prevalence of HU was 16.3% (95% CI: 15.4–17.1). Table 1 shows the prevalence of HU according to the patients’ socio-demographic characteristics. The risk of developing HU was lower in women (aOR: 0.73 [95% CI: 0.64–0.84]) and increased with age (aOR: 1.02 [95% CI: 1.02–1.03]). The risk of HU was lower in rural habitats as against urban habitats (aOR: 0.82 [95% CI: 0.68–0.97]), and in patients with primary education (aOR: 0.80 [95% CI: 0.64–0.99]) and with higher education (aOR: 0.70 [95% CI: 0.51–0.96]) as against patients with lower educational level.

In total, 35.4% (95% CI: 32.6–38.3%) of patients with HU presented some form of sTOD. The sTOD which had the highest risk of HU was left ventricular hypertrophy (LVH) (aOR: 1.36 [95% CI: 1.00–1.83]) (Table 2).

A total of 23.9% of patients with HU presented CVD. A significantly higher prevalence of HU was observed in patients with AFIB (aOR: 1.53 [95% CI: 1.19–1.98]) (Table 2). The risk of HU was also higher in patients with kidney injuries, both in eGFR < 60 mL/min (aOR: 3.01 [95% CI: 2.40–3.78]) and in albuminuria (aOR: 1.43 [95% CI: 1.14–1.78]).

The model best-fitting model (Figure 1) includes the following variables: age and sex, abdominal obesity, HTN, HF, AFIB, albuminuria, glomerular filtration rate < 60 mL/min, and use of thiazides. It can be seen that a pathological glomerular filtration, HF, abdominal obesity, HTN, use of thiazides, and albuminuria were associated with a higher risk of HU. On the other hand, being female and being actively employed were the only variables which were found to be protective factors.

## 4. Discussion

The results presented here, from a sample of 6927 patients recruited consecutively in PC, show that the prevalence of HU is 16.3%, and that the risk of HU increases in patients with both forms of kidney disease (albuminuria and glomerular filtration rate < 60 mL/min), HF, AFIB, and in the presence of CVRF such as HTN and abdominal obesity, as well as the use of thiazides. These associations have been described in various studies independently, although this is the first time that all the clinical and epidemiological variables are analyzed at the same time in a contemporary PC cohort. This has allowed us to analyze the risk of HU associated simultaneously with other variables, in order to be able to make the necessary adjustments in a multiple analysis.

The HU prevalence (16.3%) found in this study is slightly higher than other works that used the same cut-off points in neighbouring countries (13.9%) [28], or even in Asian cohorts (13.2% and 13.5%) [29,30]. Our sample shows a higher prevalence because it is a clinical practice sample, with a higher prevalence of CVRF and CVD than other population samples, as these three studies. In other ways, compared to these studies, we observed a higher prevalence in males and an increasing rate with age. This is usually because the uricemia levels increase in men aged 20 to 65 and in women older than 60; this explains that the prevalence of HU is higher in men when we analyze younger samples, and higher in women when we analyze older samples [31]. In view of this, we should consider whether, in order to define the HU as a risk factor, different cut-off points should be used not only by sex but also by age groups for both sexes. HU could be a secondary association with insulin resistance, dyslipidemia, and obesity, which are associated with an inflammatory condition state. However, uric acid is part of the body’s antioxidant defences, and is still questioned if higher levels of uric acid are a protective response to damage or a cause of disease.

Our results, obtained from a cross-sectional analysis of the entry visit of the patients of the IBERICAN study, do not enable us to establish the causal link of these CVRF and HU. Nevertheless, these results reinforce the association with obesity [32], HTN [12,33], DM [34,35], MetS [36], and even a more atherogenic lipid profile [37]. After a slow reading of these results, it is not clear if the HU is a consequence of obesity and after favours the development of HTN, hypercholesterolemia, and DM [38]. This could be related to an increased activity of xanthine oxidase in obese patients [39], but there is not enough evidence in the literature to answer this question.

This study also allowed us to analyze the simultaneous coexistence of sTOD with CVRF and CVD. It has been observed that both albuminuria and LVH have a higher risk of associated HU. This is important because both lesions are closely linked with HTN [33,40,41], but also in patients with metabolic disorders such as obesity and DM [42,43]. Something similar happens to those CVD cases associated with a higher risk of HU, such as AFIB and HF, both directly related to HTN [44]. HU was also associated with a worse prognosis in patients after acute coronary disease [23,45]. Finally, the relationship between kidney disease and HU has been confirmed, which has already been described by other authors [46,47]. This relationship has a two-way effect between the two of them [48], where both the activation of inflammation [49] and the activation of the renin–angiotensin system [50] may be involved. All this evidence might reinforce the hypothesis stated above on the role of HU in the cardiovascular continuum, from the CVRF to CVD: HU is an early marker of CVR closely linked to obesity, since the pro-inflammatory role of uric acid has been described [51], and is considered one of the pathophysiological bases of atherosclerosis [52].

From a practical point of view, HU appears to be a risk marker that reflects the oxidative stress associated with a higher CVR, from the first stages to recurrent events, and the development of kidney disease and CVD. In the absence of treatments that improve the CV prognosis with the reduction of uric acid levels, identifying patients with higher levels of uric acid could be useful to identify individuals with a higher associated CVR, and who could benefit from a better control of CVRF, giving priority to obesity, in order to improve the CV prognosis. In parallel, a careful use of thiazides should be considered in patients with HTN, since the results of our multivariate analysis have shown that both affect the risk of HU. So, taking into account that it is a first-line drug for HTN [27] and a known cause of HU [8], and a known cause of HU, the avoidance of taking thiazides seems to be a reasonable strategy for preventing HU.

### Strengths and Limitations

We believe that the results presented here are statistically robust and consistent with the available literature on this topic; in addition, they allow for a better understanding of the characteristics of the patients with HU in a contemporary cohort. The sample size is sufficient to analyze which CVRF are associated with HU, as well as the adjustment variables necessary for this study. However, our research also has limitations, some of which have been discussed in previous works [25,53], such as the voluntary participation of the investigators or the use of physical examination devices available in the practice with no regular calibration. In this sense, the information obtained in our analysis may not represent the global prevalence of HU in our country, but our main objective was to analyze the relationship between HU and other clinical situations related to cardiovascular risk including CVRF, TOD, and CVD. Despite these limitations, the authors consider it not to have influenced our findings.

Moreover, the cross-sectional design of the study does not allow us to establish causal relationships, although the aim of this study was to describe the patients’ characteristics and, therefore, its design is valid.

Lastly, in this research, the uric acid-lowering treatment of the patients was not recorded; therefore, it could not be included in the adjustment models. In this sense, it is possible that some patients classified as normouricemic would actually be HU. Even keeping that in mind, our results show the best situation, and by avoiding this bias, we would obtain more strength associations as we observed. Moreover, our results are similar to those of other authors published before, and they are biologically plausible. For these reasons, authors consider our findings answer our main research question.

## 5. Conclusions

Given the foregoing findings, it can be concluded that the risk of HU is greater in men, in the presence of CVRF such as HTN and abdominal obesity, and with the coexistence of LVH, AFIB, HF, and any form of kidney injury. These associations raise the possibility that the HU forms part of the early stages of the cardiovascular continuum, but it needs observational studies that confirm this hypothesis. However, it remains to be seen whether it is a cardiovascular risk factor and a possible therapeutic target to modify the CV prognosis, or a reflection of the existence of other metabolic disorders and of the oxidative stress characteristic of abdominal obesity which are mainly responsible for increasing the CV risk in this group of patients. If we can confirm this association, its presence in the early phases of the cardiovascular continuum makes it necessary to pay more attention in primary care to patients classified as low or moderate cardiovascular risk who have increased uric acid levels, since it could lead to reassessment of CVR.

## Figures and Tables

**Figure 1 jcm-12-00050-f001:**
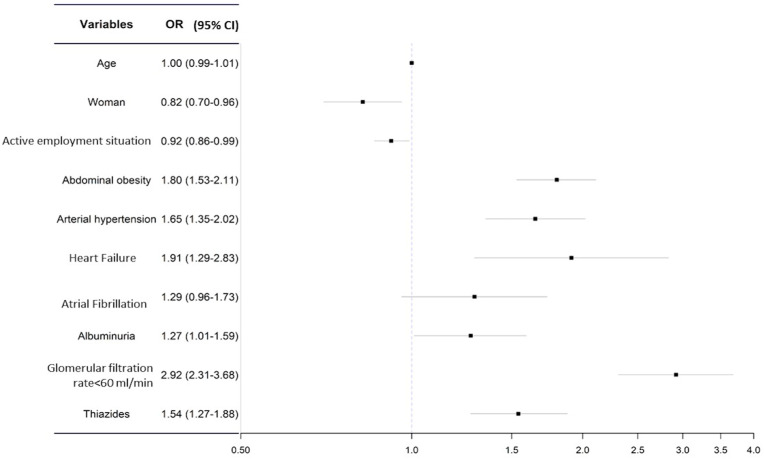
Multivariate analysis for the risk of hyperuricemia.

**Table 1 jcm-12-00050-t001:** Distribution in the prevalence of hyperuricemia according to various socio-demographic variables.

Variable		N	n	%	OR 95% CI	aOR 95% CI
Sex	Man	3188	597	18.7	0.72 [0.63–0.81]	0.73 [0.64–0.84]
	Female	3739	529	14.1		
Race	White	6678	1097	16.4		
	Others	249	29	11.6	0.67 [0.45–0.99]	0.78 [0.52–1.16]
Habitat	Urban	4061	683	16.8		
	Semi-urban	1474	237	16.1	0.95 [0.81–1.11]	0.97 [0.82–1.15]
	Rural	1390	206	14.8	0.86 [0.73–1.02]	0.82 [0.68–0.97]
Education	Uneducated	627	136	21.7		
	Primary	3814	622	16.3	0.70 [0.57–0.87]	0.80 [0.64–0.99]
	Secondary	1561	251	16.1	0.69 [0.55–0.87]	0.88 [0.67–1.14]
	Tertiary	925	117	12.6	0.52 [0.40–0.69]	0.70 [0.51–0.96]
Employment	Employed	2914	404	13.9		
	Unemployed	569	69	12.1	0.86 [0.65–1.13]	0.88 [0.66–1.17]
	Retired	2485	514	20.7	1.62 [1.04–1.87]	0.93 [0.76–1.14]
	Student	76	6	7.9	0.53 [0.23–1.23]	0.95 [0.40–2.24]
	Domestic chores	861	131	15.2	1.11 [0.90–1.38]	0.92 [0.71–1.18]
Salary (EUR/year)	<18,000	2920	490	16.8		
	≥18,000	4007	636	15.9	0.94 [0.82–1.06]	1.01 [0.87–1.16]
Age, mean (SD)	HU	62.7 (13.3)			1.02 [1.02–1.03]	1.02 [1.02–1.03]
	No HU	57.9 (14.7)				

OR = odds ratio; 95% CI = 95% confidence interval; aOR = odds ratio adjusted for sex, race, habitat, education level, employment, salary, and age. HU = hyperuricemia. Age is presented as mean and standard deviation (SD).

**Table 2 jcm-12-00050-t002:** Distribution in the prevalence of hyperuricemia according to cardiovascular risk factors, target organ damage, and history of cardiovascular disease.

Variable			N	n	%	OR 95% CI	aOR 95% CI
CVRF	Abdominal Obesity	No	4691	573	12.2	2.25 [1.97–2.58]	1.56 [1.09–2.24]
		Yes	2074	513	24.7		
	Obesity	No	4445	531	11.9	2.20 [1.92–2.52]	1.29 [0.90–1.84]
		Yes	2320	555	23.9		
	HTN	No	3469	340	9.8	2.31 [1.97–2.70]	1.95 [1.66–2.30]
		Yes	3292	746	22.7		
	Hypercholesterolemia	No	3319	422	12.7	1.32 [1.15–1.52]	1.19 [1.03–1.37]
		Yes	3443	664	19.3		
	Sedentary	No	4795	713	14.9	1.31 [1.14–1.51]	1.12 [0.97–1.30]
		Yes	1970	373	18.9		
	DM	No	5406	804	14.9	1.19 [1.02–1.39]	0.93 [0.79–1.10]
		Yes	1359	282	20.8		
	Smoking	No	5591	928	16.6	0.91 [0.75–1.10]	0.94 [0.79–1.15]
		Yes	1174	158	13.5		
TOD	PP > 60	No	5602	829	14.8	1.13 [0.94–1.36]	1.1 [0.92–1.32]
		Yes	1163	257	22.1		
	Left ventricular hypertrophy	No	6516	1022	15.7	1.44 [1.07–1.94]	1.36 [1.00–1.83]
		Yes	249	64	25.7		
	ABI < 0.9	No	6636	1060	16	1.14 [0.73–1.77]	1.06 [0.68–1.66]
		Yes	129	26	20.2		
History of cardiovascular disease	Stroke	No	6491	1030	15.9	1.10 [0.81–1.5]	1.00 [0.73–1.37]
		Yes	274	56	20.4		
	Ischemic heart disease	No	6277	996	15.9	0.96 [0.75–1.22]	0.89 [0.69–1.14]
		Yes	488	90	18.4		
	Atrial fibrillation	No	6386	973	15.2	1.76 [1.39–2.23]	1.53 [1.19–1.98]
		Yes	379	113	29.8		
	Heart failure	No	6582	1021	15.5	2.31 [1.68–3.18]	2.02 [1.44–2.83]
		Yes	183	65	35.5		
	Peripheral Artery Disease	No	6548	1045	16	0.98 [0.69–1.39]	0.88 [0.62–1.27]
		Yes	217	41	18.9		
Kidney disease	eGFR < 60 mL/min	No	6212	859	13.8	3.66 [3.01–4.46]	3.01 [2.40–3.78]
		Yes	548	226	41.2		
	Albuminuria	No	4500	687	15.3	1.70 [1.38–2.11]	1.43 [1.14–1.78]
		Yes	546	143	26.2		

OR = adjusted for sex, race, habitat, education level, employment, salary, and age; 95% CI = 95% confidence interval; aOR = adjusted for sex, race, habitat, education level, employment, salary, and age plus the variables included in its group: CVRF, TOD or CVD. CVRF: cardiovascular risk factors; HTN: hypertension; DM: diabetes mellitus; TOD: target organ damage; PP: pulse pressure; ABI: ankle-brachial index.

## Data Availability

Data sharing is not applicable to this study.

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
