# Peer review of "Prevalence of Hyperuricemia and Its Association with Cardiovascular Risk Factors and Subclinical Target Organ Damage"

_jcm, 2022, doi:10.3390/jcm12010050_

Round 1
Reviewer 1 Report
Dear Editor,
I carefully read the manuscript "Prevalence of hyperuricemia and its association with cardiovascular risk factors and subclinical target organ damage" by Antelo-Pais et al. that is really very interesting.
My comments and suggestions for the authors are the following:
- The manuscript is not very balanced in its parts. In particular, the authors should highly consider dwelling more on the rationale of their study.
- Line 135: The authors should specify at what level of p-value the statistical significance was set.
- Line 137: A flow-chart detailing the process of selection of the population sample should be included here.
- Table 2: The authors should report a part the prevalence of hypertriglyceridemia and hypercholesterolemia in the sample. "Dyslipidemia" is an obsolete diagnosis in lipidology, today.
- The limitations of the study should be further and more deeply discussed by the authors.
- In their manuscript, the authors should refer also to the findings from the well-known Brisighella Heart Study (i.e., doi: 10.1038/s41598-018-29955-w and doi: 10.1016/j.ijcard.2018.03.077).
- Moreover, the authors should highly consider referring to doi: 10.3389/fcvm.2022.978366 and doi: 10.1097/HJH.0000000000002600.
- English language needs to be carefully revised and improved.
Author Response
Dr. Emmanuel Andrès
Prof. Dr. Michael G. Hennerici
Editors Journal of Clinical Medicine
On behalf of my co-authors, thank you for your recommendations regarding our article, "Prevalence of hyperuricemia and its association with cardiovascular risk factors and subclinical target organ damage” MANUSCRIPT: jcm-2062728
We would also like to thank the referees for taking time to read our manuscript and provide constructive feedback. We have addressed the comments as explained in a point-by-point manner below, and the revisions to the manuscript are shown by tracked changes.
We hope that these revisions are satisfactory to you and the referees.
Reviewer#1
- The manuscript is not very balanced in its parts. In particular, the authors should highly consider dwelling more on the rationale of their study.
We have completed the introduction to explain better the rationale of our manuscript.
- Line 135: The authors should specify at what level of p-value the statistical significance was set.
We thank to the reviewer for the observation, and we have included in the methods section a clarification: We consider statistically significant a p-value<0.05.
- Line 137: A flow-chart detailing the process of selection of the population sample should be included here.
The patients were recruited consecutively in the clinical practice. Each researcher selected all the patients from their clinical practice the next day after authorities’ authorization. And these researchers selected at least the first 10-15 patients to complete their sample. We understand that in this type of recruitment it is not necessary to insert any flow-chart, but we added the next one to the reviewer confirm it.
- Table 2: The authors should report a part the prevalence of hypertriglyceridemia and hypercholesterolemia in the sample. "Dyslipidemia" is an obsolete diagnosis in lipidology, today.
We agree with the reviewer and change the term “Dyslipidemia” by “Hypercholesterolemia” because in fact we only know which patients had high levels of cholesterol.
- The limitations of the study should be further and more deeply discussed by the authors.
We have completed this section with new comments about the different possible bias and their impact on our results.
- In their manuscript, the authors should refer also to the findings from the well-known Brisighella Heart Study (i.e., doi: 10.1038/s41598-018-29955-w and doi: 10.1016/j.ijcard.2018.03.077).
- Moreover, the authors should highly consider referring to doi: 10.3389/fcvm.2022.978366 and doi: 10.1097/HJH.0000000000002600.
We thank the reviewer for these recommendations and we have included them.
- English language needs to be carefully revised and improved.
We have reviewed our English spelling

Reviewer 2 Report
This is an observational study on 6927 patients included in the IBERICAN study, consecutively recruited from the National Health System, focused on the association between hyperuricemia and cardiovascular risk factors, subclinical target organ damages and cardiovascular diseases.
The Authors found that hyperuricemia was associated with hypertension, low glomerular filtration rate and with left ventricular hypertrophy, atrial fibrillation and heart failure. These associations had also been reported by previous studies.
The topic is interesting, and the study well conducted, on an adequate number of patients.
I have some concerns:
1. In the Discussion (lines 199-200), the Authors stated that “it can be posed that, with our results, the HU plays its role in the earliest phases of the cardiovascular continuum”. This concept needs to be clarified. In my opinion, only the association between hyperuricemia and other cardiovascular risk factors can be highlighted with the results of the study. Moreover, previous studies affirm the association between hyperuricemia and the extension of obstructive coronary artery disease and also between hyperuricemia and worse clinical outcomes after acute events such as acute coronary syndromes. Consequently, it is possible that uric acid may also play a role in later than the initial stage of cardiovascular disease.
2. No significant association between hyperuricemia and ischemic heart disease was found in the study. Is it possible to make a distinction between chronic coronary syndromes and acute coronary syndromes?
3. Also, are there data regarding cardiovascular mortality and all-cause mortality?
4. There are some typos in the abstract and in the text.
Author Response
Reviewer#2
This is an observational study on 6927 patients included in the IBERICAN study, consecutively recruited from the National Health System, focused on the association between hyperuricemia and cardiovascular risk factors, subclinical target organ damages and cardiovascular diseases.
The Authors found that hyperuricemia was associated with hypertension, low glomerular filtration rate and with left ventricular hypertrophy, atrial fibrillation and heart failure. These associations had also been reported by previous studies.
The topic is interesting, and the study well conducted, on an adequate number of patients.
We thank the reviewer for their positive feedback and appreciate their insightful comments below.
I have some concerns:
- In the Discussion (lines 199-200), the Authors stated that “it can be posed that, with our results, the HU plays its role in the earliest phases of the cardiovascular continuum”. This concept needs to be clarified. In my opinion, only the association between hyperuricemia and other cardiovascular risk factors can be highlighted with the results of the study. Moreover, previous studies affirm the association between hyperuricemia and the extension of obstructive coronary artery disease and also between hyperuricemia and worse clinical outcomes after acute events such as acute coronary syndromes. Consequently, it is possible that uric acid may also play a role in later than the initial stage of cardiovascular disease.
We agree with the reviewer that the HU is associated with CVRF but also with a worst prognosis in patients after acute coronary syndrome. We have changed some sentences in the discussion in this regard, and also, we have deleted the sentence that the referee indicates in his comments.
- No significant association between hyperuricemia and ischemic heart disease was found in the study. Is it possible to make a distinction between chronic coronary syndromes and acute coronary syndromes?
We show our results of the transversal analyses of the inclusion visit of IBERICAN, and we only have information about the previous diagnosis of coronary disease. Because of this, we only can refer the chronic coronary syndromes. In the future, when we have the follow-up of the patients, we will know about the acute coronary syndromes.
- Also, are there data regarding cardiovascular mortality and all-cause mortality?
As in our response to the previous question, in this moment we only have the personal history of cardiovascular disease. With the follow-up information we will know all types of mortality.
- There are some typos in the abstract and in the text.
We have reviewed our English spelling

Reviewer 3 Report
To the Authors
Major Issues
This is a retrospective observational study using the dataset of IBERICAN study. The total number of patients is 6,927. The authors tried to elucidate the association between HU and CVRF, sTOD and CVD.
1. There are several objectives in this study. It would be desirable to narrow the target of the study since it would make this study easier to be understood by the readers. They are to show the co-existence of HU and cardiovascular problems (CVP) including CVRF, sTOD and CVD, to show the cause-and-effect relation between HU and CVP, to show the prevalence of HU in the population, to show that HU is the early surrogate of CVP, and so on.
2. An appropriate statistical model should be selected that can validate the intention or hypothesis of this study. The target value of the multivariate model seems to be HU as shown in Figure 1. This model could show which parameters are the risk of developing HU, from among age, gender, obesity, HTN, HF and so on. Therefore, HU is treated as the effect of CVP. It might be contrary to the intention of the authors.
3. If the authors intend to prove that HU is the cause of CVP, some kind of intervention, clinically or statistically, would be needed. In this case, the propensity score method might work. After equalizing the risks of developing CVP, such as HTN or HF, with propensity score matching between paired datasets, the authors would be able to show if HU is a possible risk of developing CVP or not.
4. If HU is a risk of developing CVP, prevalence of ischemic heart disease and peripheral artery disease, both of which are main CVP, should be higher in the HU group. However, the result of the survey was not. The reason of this should be explained.
Minor issues
1. The level of uric acid is strongly influenced by renal function. Since patients with impaired renal function is only 548, it might be a choice to perform a sub-analysis using the dataset without patients of low renal function.
2. Line 131 in page 3. What type of adjustment was used between the interactive confounding factors?
3. Line 168 in page 5. What is meant by protective factors?
4. Line 175 in page 5. It is written as “both forms of kidney disease”. Which renal disease?
5. Line 194-228 in page 6. The outline of the explanation for the association between HU and CVP or other diseases is difficult to understand with too many assumptions and reports. It should be simplified for readers who are not so familiar with HU issues.
Author Response
Major Issues
This is a retrospective observational study using the dataset of IBERICAN study. The total number of patients is 6,927. The authors tried to elucidate the association between HU and CVRF, sTOD and CVD.
First of all, we want to thank to the reviewer his feedback and comments, that we are going to answer in the next letter in the best way possible.
- There are several objectives in this study. It would be desirable to narrow the target of the study since it would make this study easier to be understood by the readers. They are to show the co-existence of HU and cardiovascular problems (CVP) including CVRF, sTOD and CVD, to show the cause-and-effect relation between HU and CVP, to show the prevalence of HU in the population, to show that HU is the early surrogate of CVP, and so on.
We want to demonstrate that the HU is associated at the same time with the CVRF, TOD and CVD. In really, maybe it is a common cause of all the continuum cardiovascular as we have exposure in the introduction of our manuscript, and the interpretation of this situation in the practice will be that the patient with high levels of uricemia has a higher CV risk. This is very important in the patients with low to moderate estimated CV risk because the presence of HU could increase the estimated risk and then the physician must change the treatment to a more intensive strategy.
Now, the available data show us the association between HU and some CVRF and CVD and also, with the prognosis after a CV event. However, no author to date has performed an analysis at once for all CVRF, CVD, and TOD with HU. Our results show that is very usual the association with only some CVRF, TOD, and AF and HF as CVD. In this sense, the CV risk would be higher in these patients, and they had a worst prognosis. For this reason, although our results must be interpreted with caution, this study may improve the follow-up of these patients and help develop new tools.
Our study only allows to analyse associations, because we have used the data in the inclusion visit of our study and we cannot conclude the cause relationship between these variables with the HU. Because of this, we show all the results in the same analyses.
- An appropriate statistical model should be selected that can validate the intention or hypothesis of this study. The target value of the multivariate model seems to be HU as shown in Figure 1. This model could show which parameters are the risk of developing HU, from among age, gender, obesity, HTN, HF and so on. Therefore, HU is treated as the effect of CVP. It might be contrary to the intention of the authors.
As we have explained in the statistical analysis section, we used unconditional logistic regression models in first step and a multivariate analysis in the second step. The results were showed in both tables and figure 1.
It is necessary to mention that HU could be secondary association with insulin resistance, dyslipidemia, and obesity, which are associated with an inflammatory condition state. However, acid uric is part of the body’s antioxidant defences, and is still questioned if higher levels of uric acid are a protective response to damage or a cause of disease. We included this comment in the line 200 in the discussion section.
- If the authors intend to prove that HU is the cause of CVP, some kind of intervention, clinically or statistically, would be needed. In this case, the propensity score method might work. After equalizing the risks of developing CVP, such as HTN or HF, with propensity score matching between paired datasets, the authors would be able to show if HU is a possible risk of developing CVP or not.
We agree that the propensity score method is the best to analyse the relationship between tow variables, for example HU as a cause of CVRF. But, as we explained in the previous comments, we only analyse the coexistence or the presence at the same time of the HU with other CVRF and CVP, we need a follow-up analysis to confirm a cause-relationship between them, and it is impossible with our analysis.
- If HU is a risk of developing CVP, prevalence of ischemic heart disease and peripheral artery disease, both of which are main CVP, should be higher in the HU group. However, the result of the survey was not. The reason of this should be explained.
Exactly, these are the results. As we explain in the previous comment (#3) our show the results of a cross-sectional analysis, and maybe we cannot observe a correct relationship between variables. We only show what were the most frequent associations and in what clínica situations (CVRF, TOD and CVD) is more frequent or has more risk to present at the same time HU.
Minor issues
- The level of uric acid is strongly influenced by renal function. Since patients with impaired renal function is only 548, it might be a choice to perform a sub-analysis using the dataset without patients of low renal function.
We agree that maybe it would be interesting to analyse the relationship between HU and other variables in patients without renal disease. But the objective of our study is to analyse this relationship in the sample IBERICAN, a sample descripted in previous works and with their characteristics descripted yet.
Probably, the reviewer indicated us a new line of work, but we think that this is a different objective that we showed in our manuscript.
- Line 131 in page 3. What type of adjustment was used between the interactive confounding factors?
We have used a multivariate logistic regression model adjusted by confounding variables as fixed effects described in statistical analysis.
- Line 168 in page 5. What is meant by protective factors?
In the line 177 of this page, we refer the protective factor to explain the factor in what the HU has a lowest prevalence or with minus frequency.
- Line 175 in page 5. It is written as “both forms of kidney disease”. Which renal disease?
Along the manuscript we use two types of renal disease: glomerular filtration rate <60ml/min and albuminuria. When we have used the term “both forms of kidney disease” we refer about these both types of renal disease.
- Line 194-228 in page 6. The outline of the explanation for the association between HU and CVP or other diseases is difficult to understand with too many assumptions and reports. It should be simplified for readers who are not so familiar with HU issues.
We have modified these both paragraphs to explain more clearly the relationship between HU with other CVRF and to explain our hypotheses in our manuscript.

Reviewer 4 Report
The manuscript by Paula Antelo-Pais et al. entitled “Prevalence of hyperuricemia and its association with cardiovascular risk factors and subclinical target organ damage” to analyse which CVRF, sTOD, and CVD are associated with the presence of HU in patients seen in primary care practices.
I read with great interest this paper. The abstract summarizes the general significance of the manuscript, and the article leads some evidence to such point. The methods section and the inclusion / exclusion criteria are well explained. Moreover, the conclusions are supported by the results.
In conclusion, the manuscript can be accepted.
Author Response
On behalf of all the co-authors of the work, we welcome your comments, and we are very pleased that reading our work has been to your complete satisfaction.
Best regards
Round 2
Reviewer 2 Report
The Authors answered all my questions.
Author Response
The authors would like to thank the reviewer for all his comments on our paper, which has greatly improved its quality.
Reviewer 3 Report
To the Authors
Major Issues
The revised article is more focused on the facts that HU coexist with other diseases, such as HTN, obesity, renal disease and so on. Therefore, the story of the article is clearer for the readers. At the same time, however, the content of this article has become the assertion of facts that have been already reported.
The strength of this article is that it is based on the sound dataset composed of well-controlled patients. Since the number of records is high, the dataset should include more information regarding HU than those revealed by the analysis conducted so far.
Minor Issues
1. Page 5 line 182: It is still unclear which renal diseases are indicated by “both forms of kidney disease”.
2. Page 6 line 191: … prevalence of CVRF and CVD that other population…. . Is “that” a mistype of “than” ?
3. Page 6 line 200: … However, acid uric is part of the … . Is “acid uric” a mistype of “uric acid” ?
Author Response
Reviewer#3
Major Issues
The revised article is more focused on the facts that HU coexist with other diseases, such as HTN, obesity, renal disease and so on. Therefore, the story of the article is clearer for the readers. At the same time, however, the content of this article has become the assertion of facts that have been already reported.
The strength of this article is that it is based on the sound dataset composed of well-controlled patients. Since the number of records is high, the dataset should include more information regarding HU than those revealed by the analysis conducted so far.
We thank to the reviewer these comments, and we would like to answer them. As we have comment in our manuscript, other authors show results in the same way with associations between HU and other CVRF and CVD, but all of them did these analyses using some CVRF or the CVD without other variables. Our study let us to do all these analyses at the same time using al the CVRF, TODs and CVD and also let us get consistent conclusions about the correct associations.
We think that this is a very important stronghold of the IBERICAN study in general and about our manuscript, because of this, we decided to publish it in your journal.
Minor Issues
- Page 5 line 182: It is still unclear which renal diseases are indicated by “both forms of kidney disease”.
We have added a clarification: “(albuminuria and glomerular filtration rate<60ml/min),”
- Page 6 line 191: … prevalence of CVRF and CVD that other population…. Is “that” a mistype of “than”?
We are sorry, it was a mistake, the correct sentence is: The HU prevalence found (16.3%) is slightly higher than other works that used the same …
- Page 6 line 200: … However, acid uric is part of the …. Is “acid uric” a mistype of “uric acid”?
Exactly: the correct form is uric acid.
